# Novel Alumina Matrix Composites Reinforced with MAX Phases—Microstructure Analysis and Mechanical Properties

**DOI:** 10.3390/ma15196909

**Published:** 2022-10-05

**Authors:** Mateusz Petrus, Jaroslaw Wozniak, Tomasz Cygan, Wojciech Pawlak, Andrzej Olszyna

**Affiliations:** 1Faculty of Material Science and Engineering, Warsaw University of Technology, ul. Wołoska 141, 02-507 Warsaw, Poland; 2Faculty of Mechanical Engineering, Lodz University of Technology, Stefanowskiego 1/15 St, 90-924 Lodz, Poland

**Keywords:** sintering, composites, mechanical properties, Al_2_O_3_, MAX phases

## Abstract

This article describes the manufacturing of alumina composites with the addition of titanium aluminum carbide Ti_3_AlC_2_, known as MAX phases. The composites were obtained by the powder metallurgy technique with three types of mill (horizontal mill, attritor mill, and planetary mill), and were consolidated with the use of the Spark Plasma Sintering method at 1400 °C, with dwelling time 10 min. The influence of the Ti_3_AlC_2_ MAX phase addition on the microstructure and mechanical properties of the obtained composites was analyzed. The structure of the MAX phase after the sintering process was also investigated. The chemical composition and phase composition analysis showed that the Ti_3_AlC_2_ addition preserved its structure after the sintering process. The increase in fracture toughness for all series of composites has been noted (over 20% compared to reference samples). Detailed stereological analysis of the obtained microstructures also could determine the influence of the applied mill on the homogeneity of the final microstructure and the properties of obtained composites.

## 1. Introduction

Alumina is one of the most commonly used ceramics. This is due to its unique properties, such as high hardness, good thermal and chemical stability, high strength, high young modulus and low price [1,2]. These advantages allow monolithic ceramics to be used in electronics, power generation, cutting tools, aerospace, military or biomedical [3,4]. However, one of the main disadvantages, which limits the potential use, is low fracture toughness. The method to improve the fracture toughness of ceramics is the production and development of new multiphase composites [5,6]. One of the most interesting groups of materials, characterized by a combination of the properties of ceramics and metals that can potentially be used as a phase reinforcing the ceramic matrix, is the MAX phases [7,8,9,10,11]. The MAX phases constitute a large group of anisotropic crystalline materials. Their name reflects their composition M_n+1_AX_n_ where M—light transition metal, e.g., Ti, Nb, V; A—metals from group 13 or 14, whereas X is carbon or nitrogen [12,13]. The MAX phases exhibit a layered structure with a metallic layer between layers M–X. Since the discovery of the first MAX phases in 1967 by Nowotny’s team, more than 60 different, thermodynamically stable MAX phases have been developed [10,14]. Due to the presence of covalent–metallic–ionic bonds, the MAX phases are characterized, like ceramics, by high hardness, high strength properties, good thermal properties and like metals, they display good electrical and thermal conductivity, good thermal shock, high fracture toughness and are relatively easily mechanically machined [15,16,17].

In recent years, there has been a growing interest in using MAX phases as an addition to the ceramic matrix to improve ceramic composites’ sinterability and mechanical properties. Titanium aluminum carbide (Ti_3_AlC_2_) is a typical member of the MAX phases with a density of 4.2 g/cm^3^, fracture toughness of 9.1 MPa·m^0.5^ and flexural strength over 550 MPa [18]. However, Ti_3_AlC_2_ is unstable and decomposes above 1300 °C. Nayebi et al. [19] used the addition of the Ti_3_AlC_2_ phase to lower the sintering temperature and produce TiB_2_-based composites using the SPS sintering technology. During the sintering, the authors confirmed the decomposition of the Ti_3_AlC_2_ phase with the production of Al, Ti, and TiC_0.67_. Similar studies have been conducted for ZrB_2_ [20]. The decomposition of Ti_3_AlC_2_ during sintering promoted the reactive sintering mechanism by forming several in-situ synthesized secondary phases. The aforementioned phases can be categorized into three main groups: Al-, Ti- and Zr-rich reinforcement, such as Al_2_O_3_, AlOC, Al_2_OC, TiB_2_, and ZrC. The presence of TiC, which is a decomposition product, was also confirmed in the microstructure. The reactive sintering mechanism in the B_4_C-Ti_3_AlC_2_ systems was also noted [21,22]. These studies also indicated the problem of TiC agglomerates formation, which has not been solved yet. Thus far, the research on the Ti_3_AlC_2_ phase has focused mainly on non-oxide ceramics, which require the use of temperatures higher than the MAX phase thermal stability [23]. They used the MAX phase decomposition to form aluminum and TiC, which reacted with the sintered material. These studies also suggest a problem with the agglomeration of the MAX phases during the production of composites, but thus far, it has not been thoroughly tested and described. However, there are no studies on the use of the Ti_3_AlC_2_ phase as an additive to aluminum oxide, the sintering temperature of which is much lower than that of non-oxide ceramics and amounts to 1300–1400 °C [24]. It should prevent the decomposition of the MAX phases, thus enabling the development of a new family of ceramic composites.

This article aims to produce alumina matrix composites reinforced with Ti_3_AlC_2_. The MAX phases were obtained through the Self-Propagating High-Temperature Synthesis (SHS) using the Spark Plasma Sintering Method (SPS) from pure elements. The effect of the addition of Ti_3_AlC_2_ MAX phase and the type of applied mill type on the microstructure and mechanical properties of the composites were investigated.

## 2. Materials and Methods

The commercial powders of titanium aluminum, synthetic graphite and alumina were used in the process of producing the Al_2_O_3_—Ti_3_AlC_2_ composites (Table 1).

To prepare MAX phases, the titanium, aluminum, and carbon powders were wet blended in propan-2-ol (no. 1759, Stanlab, Lublin, Poland) with the use of a ball-type mill (Fritsch Pulverisette, Fritsch, Idar-Oberstein, Germany). After that, the obtained powder mixture was dried and sieved (# 300 µm). The molar ratio of Ti:Al:C = 3:1:1.9 was applied. The composition of the mixture was developed on the basis of optimization tests, taking into account the phase purity of the obtained products. The reactive synthesis process of synthesis the Ti_3_AlC_2_ phase was performed using the Spark Plasma Sintering technique (FCT Systeme GMBH, Effelder Rauenstein, Germany). The following parameters were applied: temperature: 1300 °C, heating rate: 250 °C/min, vacuum atmosphere (*p* = 5 × 10^−2^ mbar). The MAX phase was ground with an automatic mortar grinder (Retsch KM100, Retsch GmbH, Haan, Germany) below 45 µm (grinding bowl speed = 70 rpm, applied force 50–125 N).

The Al_2_O_3_—Ti_3_AlC_2_ composites were prepared with the powder metallurgy technique and sintered with the SPS method. First, the Al_2_O_3_-xTi_3_AlC_2_ powder mixtures (where x = 5, 10, 15, 20 wt%) were wet blended in three different ball-type mills: horizontal mill, attritor mill and planetary mill in propan-2-ol and alumina grinding balls (Nikkato, Osaka, Japan). The obtained powder was dried and manually sieved (# 300 µm). Composites were sintered with the use of SPS method. The process parameters were as follows: sintering temperature: 1400 °C, heating, and cooling rate: 250 °C/min, 4 min dwell time, 50 MPa applied pressure and vacuum (*p* = 5 × 10^−2^ mbar). As a reference sample, pure alumina sinters were prepared.

The density of the specimens was examined using the helium pycnometer (Ultrapycnometer 1000, Quantachrome Instruments, Boynton Beach, FL, USA). Hardness of the obtained composites was measured under the load of 49.05 N with the Vickers Hardness Tester (FV-700e, Future-Tech, Kawasaki, Japan). The Vickers Indentation Fracture (VIF) of the produced composites was determined based on the crack length generated at the corners of the Vickers indenterunder the load of 49.05 N. The Niihara, Morena, Hasselman equation was used. For each sample, 20 hardness measurements and 12 crack length measurements were performed. The phase composition of composites was analyzed with X-ray diffraction (XRD, PANalytical Empyrean, Malvern Panalytical, Malvern, UK), using CuKα radiation at a wavelength of 0.15406 nm. The parameters of this test were as follows: voltage: 45 kV, current: 40 mA, angular range: 10–155 deg with step 0.03. The microstructure observations were performed on a scanning electron microscope (SEM Hitachi 5500, Hitachi, Tokyo, Japan). The observations were carried out at a 20 kV accelerating voltage. The quantitative description of the microstructure was performed on binarized images using the following parameters: average equivalent diameter of matrix grains size, average equivalent diameter of agglomerates $\bar{d^{2}}$, a circularity of agglomerates (η  =  4πS/O^2^), elongation of agglomerates defined as the ratio of the maximum to the minimum Feret diameter. The homogeneity of the microstructure in terms of the agglomerate distribution of the Ti_3_AlC_2_ phase was estimated using the skeletization by the function of SKIZ (skeleton by influence zone). In this method, the fields of influence of structure elements are determined. By determining the mean equivalent diameter of these fields and subjecting it to statistical analysis, it is possible to compare the homogeneity of the structures of the tested materials. Stereological analysis was performed using the NIS ELEMENTS BR 5.30-03 software (Nikon, Tokio, Japan).

## 3. Results

### 3.1. Characterization of MAX Phases

The morphology of the obtained Ti_3_AlC_2_ MAX phase is presented in Figure 1a. The layered structure, characteristic of this group of materials, can be observed, which verifies the correct occurrence of the MAX phase synthesis process. This is also confirmed by the results of the phase composition of powders (Figure 1b). The XRD analysis exhibited the presence of Ti_3_AlC_2_ as the main phase and slight amounts of TiC, Ti_2_AlC and graphite.

### 3.2. Microstructure Analysis

Figure 2 presents the relative density of obtained Al_2_O_3_-Ti_3_AlC_2_ composites. The reference samples have a relative density of 99.3%. It can be seen that the composites obtained with the use of the planetary mill and the attritor type mill show a similar course of density changes as a function of the Ti_3_AlC_2_ MAX phase content: a decrease in the value for the 5 wt% additive, then an increase until reaching the maximum for the 15% additive and a decrease for the 20 wt% additive. For composites obtained with a horizontal mill, the decrease for 5 wt% is smaller and the subsequent increase in density with increasing Ti_3_AlC_2_ phase addition becomes linear until reaching the maximum for 20 wt%. The composite with the addition of 15 wt% of Ti_3_AlC_2_ obtained using a planetary mill shows the highest relative density.

The fracture surfaces of obtained sinters are shown in Figure 3. A presence of numerous fine pores characterizes the microstructure of the reference sample. In the case of composites with the addition of 10 wt% Ti_3_AlC_2_ phase, irrespective of the mill used, a finer grain was observed, as well as the presence of pores and voids. Comparing the fracture surfaces in terms of the mill used, we can see that the finest grains and, simultaneously, the highest amount of pores and voids occur in the composite obtained using a planetary mill. In the case of composites, there were also areas where the fracture mechanism changed from transcrystalline to intracrystalline (white arrows).

Observations of the composites microstructure also showed the presence of an additional phase with a layered structure. Its morphology corresponds to that used as the starting powder of the MAX Ti_3_AlC_2_ phase (Figure 4a). Depending on the mill used, a different degree of bonding of the observed phase with the matrix can be seen. More pores and discontinuities at the interface were observed in the composites obtained with the use of the attritor mill (Figure 4b—white arrows). In the case of composites obtained with the use of a horizontal mill, the presence of a significant amount of second phase agglomerates with sizes exceeding 100 μm was also observed (Figure 4c).

The analysis of the chemical composition of the observed layered structures (Figure 5) showed that they consist of titanium, aluminum and carbon, which corresponds to the chemical composition of the used Ti_3_AlC_2_ phases. It suggests that the MAX phases did not decompose during sintering and retained their original structure. The study of phase composition confirmed the above finding (Figure 6). XRD analysis showed that the obtained composites are mainly composed of Al_2_O_3_, Ti_3_AlC_2_, and a small amount of TiC. The presence of TiC may be related to the fouling of the starting powder of the Ti_3_AlC_2_ (Figure 1b) phase or the possibility of partial decomposition of the Ti_3_AlC_2_ phase during the sintering process.

### 3.3. Stereological Analysis

A complete stereological analysis of the microstructures of the obtained composites was carried out. Figure 7 shows the mean equivalent grain size of the produced composites. As can be seen, the addition of 5 wt% of the Ti_3_AlC_2_ phase causes a decrease in the average grain size by over 80% (2.2 µm and 0.42 µm for reference Al_2_O_3_ and Al_2_O_3_—5 wt% Ti_3_AlC_2_ sinter, respectively). A further increase in the MAX phase content does not cause a further decrease in d_2_, all produced composites, regardless of the composition and type of the mill used, show an average grain size oscillating between 0.3–0.5 µm.

Figure 8 shows the average equivalent diameter of the Ti_3_AlC_2_ phase agglomerates. In the case of composites made with a horizontal mill and an attritor mill, an increase in the average size of agglomerates was observed with the increase in Ti_3_AlC_2_ phase addition up to 15 wt% additive by nearly 80%, compared to 5 wt% additive. For the 20 wt% addition of the Ti_3_AlC_2_ phase, an insignificant decrease in the average size of the agglomerates was noted. A slightly different trend was observed for composites obtained using a planetary mill. In the analyzed range, much smaller average sizes of the formed agglomerates were noted. The increase in size with the increase in the Ti_3_AlC_2_ phase was insignificant and amounted to a maximum of 20% for the composite with the addition of 20 wt%.

The aspect ratio of the formed agglomerates was also analyzed (Figure 8). The circularity of the Ti_3_AlC_2_ phase agglomerates in all the obtained composites oscillates around 0.8 and shows no changes depending on the amount of addition of the MAX phase (Figure 9a). The elongation behaves similarly, which for all composites in the entire analyzed range fluctuates around 1.5, which indicates a slight elongation of the agglomerates (Figure 9b).

Figure 10 shows the coefficients of variability of the surface area of cells resulting from SKIZ tessellation (images of macro-regions analyzed by SKIZ tessellation were presented in Appendix A). In the case of composites obtained with the use of the horizontal and the attritor mill, an increase in the coefficient of variation is observed with the addition of the Ti_3_AlC_2_ phase. In contrast, in the case of composites from the attritor, this increase is more significant. This proves the decrease in the homogeneity of the Ti_3_AlC_2_ phase distribution in the discussed composites. In the case of composites obtained with a planetary mill, no decrease in the homogeneity of the microstructure as a function of the composition content was observed. In the entire analyzed range, these composites were characterized by the lowest CV coefficient and, thus, the highest degree of homogeneity of the microstructure.

### 3.4. Mechanical Properties

Figure 11 shows the mechanical properties of the obtained composites. In the case of hardness, we can see that the reference sample is characterized by a high hardness value of 1843 HV. The hardness of composites, in most cases, achieved a lower value than the reference samples. Only samples containing 5, 10 and 15 wt% MAX phases mixed in a horizontal mill and a 15 wt% MAX phase sample mixed in a planetary mill achieved higher hardness values than the unreinforced sinter. A similar trend can be seen for all series of composites. The hardness value increases with an increase in the content of the reinforcing phase to about 15 wt%, and then, a slight decrease is observed. The fracture toughness of obtained composites is shown in Figure 11b. All produced composites show higher fracture toughness compared to the reference sample. The highest value measured was 5.7 MPa·m^0.5^ for the composite with the addition of 15 wt% Ti_3_AlC_2_ obtained using an attritor mill.

## 4. Discussion

The SEM observations of the microstructure, the chemical composition analysis and the phase composition analysis showed that it is possible to obtain composites based on aluminum oxide reinforced with the Ti_3_AlC_2_ phase. Despite the use of the sintering temperature exceeding the temperature of the Ti_3_AlC_2_ phase decomposition, their original structure can be preserved. The small amount of TiC found during the phase composition analysis may come from both contamination of the starting powder of the MAX phase and may indicate the decomposition of a small amount of the MAX phase during the sintering process. However, no characteristic clusters of TiC grains signaled in other works were found [25], which indicates that the decomposition process has been successfully limited. It is probably related to the applied SPS sintering technology. Shortening the sintering time to a few minutes limits the decomposition process [26]. All the produced composites were characterized by a relative density above 95%, while for the 5 wt% addition of the Ti_3_AlC_2_ phase, a clear decrease in the mean value of the relative density was observed. This decrease is particularly noticeable for the series of composites obtained with the use of a horizontal and planetary mill. Similar changes were observed in the case of composites with the addition of 20 wt% Ti_3_AlC_2_ phase. Microscopic observations indicate that porosity is generated mainly within the flakes of the Ti_3_AlC_2_ phase, at the interface and between the matrix grains. As already mentioned, for composites with a small content of the Ti_3_AlC_2_ phases, the observed changes in relative density can be caused by several factors. A small amount of MAX phases may undergo thermal decomposition during sintering. The amount of phase that decomposes and the amount of products formed is small, but it can affect the calculated theoretical density and thus distort it. In addition, three different mills with different mixing energies were used during the composites manufacturing process. This may affect the different degree of contamination of composites with the dope coming from the used grinding balls (Al_2_O_3_) and the blades in the attritor (ZrO_2_). This contamination may also cause changes in theoretical density, which will affect the theoretical density of composites, and thus affect the density of the measured relative density [27,28]. The microstructure analysis also shows that, regardless of the mill used in the technological process, the Ti_3_AlC_2_ phase distributed among the matrix grains effectively reduces the growth of the Al_2_O_3_ matrix grain in relation to the reference sample. The phenomenon of limiting the grain growth in composites on a ceramic matrix, even for a small amount of additives, has been signaled in the literature [29]. The addition of the Ti_3_AlC_2_ phase also changes the nature of the cracking of the composites from intergranular to transgranular (mark by arrow at Figure 3a–d). The reference sample breaks almost completely intergranular, while with the addition of the reinforcing phase increase, the proportion of transgranular cracking increases. This proves that the cohesive forces are increased in the case of composites compared to the unreinforced sinter. Significant differences were also found between individual series of composites depending on the mill used. The highest content of transgranular cracking was observed for composites mixed in the attritor. These differences are manifested both on the macro and micro scale. In the case of composites obtained with the attritor mill, pores and voids were found at the Al_2_O_3_-Ti_3_AlC_2_ interface, which suggests poor bonding of the reinforcement phase flakes with the matrix (Figure 4b). Moreover, the type of mill used has a significant influence on the formation of agglomerates. This aspect was signaled in the literature as one of the main technological problems in the case of this type of material [30]. In the case of composites obtained with a horizontal mill and an attritor, an increase in the average size of Ti_3_AlC_2_ phase agglomerates is observed with the increase in Ti_3_AlC_2_ phase addition. At the same time, the coefficient CV_SKIZ_ increases, which indicates a decrease in the homogeneity of the obtained microstructure. The increase in the size of agglomerates in composites is not reflected in the results of density measurements. Increasing the dimensions and content of agglomerates is usually associated with a decrease in density caused by the occurrence of voids between the particles inside the agglomerates [31,32]. In the case of the produced composites, due to the similar sintering temperature of Al_2_O_3_ and Ti_3_AlC_2_, the particles of the MAX phase were consolidated inside the agglomerates (Figure 4c). Thus, they did not lower the density of the composites. A much more significant impact on the density change in individual composites has the content of impurities remaining from the starting powders and a slight decomposition of the MAX phases during the sintering process.

The analysis of the shape of the formed agglomerates shows that it does not change with the increase in the average size of the agglomerates. Composites made using a planetary mill have a completely different character. The observed increase in the average size of agglomerates is slight, while the obtained microstructures are characterized by a high degree of homogeneity, regardless of the amount of Ti_3_AlC_2_ phase addition. High mixing energy contributes to the more effective breaking of the agglomerates at this stage, which reduces their average size and effects on, increasing the homogeneity of the mixture and, consequently, increasing the homogeneity of the obtained microstructure [33].

Despite significant differences in the microstructure of the obtained composites, the observed differences in mechanical properties are lesser. In the case of hardness, the results for composites will be related to several factors. Namely, the average size of the matrix grain, the average hardness of the individual components and the degree of homogeneity of the matrix. In the case of Al_2_O_3_ and Ti_3_AlC_2_, these phases’ hardness differences are significant; the crystal hardness is 22 GPa and 2.7 GPa, respectively [34]. This suggests that the hardness of the produced composites should be much lower than indicated by the research. The high hardness of the tested composites, close to the hardness of the reference sample sintered at 1400 °C, is probably associated with a clear decrease in the average grain size of the matrix resulting from the addition of the Ti_3_AlC_2_ phase (Figure 7) and the strengthening effect caused by the presence of an additional phase at the grain boundaries. The homogeneity of the Ti_3_AlC_2_ phase distribution is also important in this case. This is confirmed by the lowest standard deviations of the averaged hardness of composites obtained with a planetary mill, which are also characterized by the highest degree of homogeneity. Fluctuations in the hardness obtained in the case of the remaining series probably result from the overlapping of the effects mentioned above, the decrease in the degree of microstructure homogeneity and an increase in the average size of agglomerates. The situation is similar in the case of fracture toughness. All produced composites are characterized by a higher VIF than the reference samples, which confirms that adding the Ti_3_AlC_2_ phase effectively increases the fracture toughness. Improving the fracture toughness of composites reinforced with MAX phases is related to their structure. Due to the alternating occurrence of metallic and ceramic layers, the propagated fracture can be bridged, can deflect causing delamination of individual layers or lower the fracture energy by plastic deformation of the metallic layers [35]. However, the fracture toughness of the tested composites, despite a significant increase, does not show significant changes depending on the used mill or the amount of reinforcement.

## 5. Conclusions

Based on the obtained results, it can be concluded that it is possible to obtain dense alumina composites reinforced with Ti_3_AlC_2_ MAX phases. SPS sintering technology effectively reduces the decomposition of the used MAX phase. In the case of the production of this type of composite, it is important to select the appropriate type of mill. The most homogeneous microstructure in terms of the size and distribution of Ti_3_AlC_2_ agglomerates was obtained using the planetary mill with the highest mixing energy. The application of the MAX phases as a reinforcing phase in the alumina allows one to obtain sinters with high hardness, similar to the hardness of pure sinter while improving the fracture toughness. The composites showed almost 20% higher fracture toughness while maintaining high hardness. Considering the obtained mechanical properties of the composites and the possibility of maintaining the MAX phase structure after the sintering process, new opportunities for producing a new family of ceramic materials appear.

## Figures and Tables

**Figure 1 materials-15-06909-f001:**
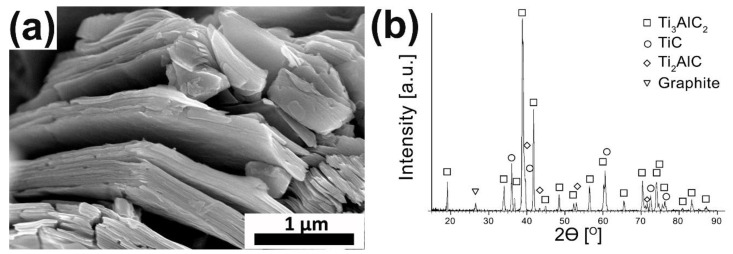
The (**a**) morphology and (**b**) XRD analysis of synthesized Ti_3_AlC_2_ MAX phase powders.

**Figure 2 materials-15-06909-f002:**
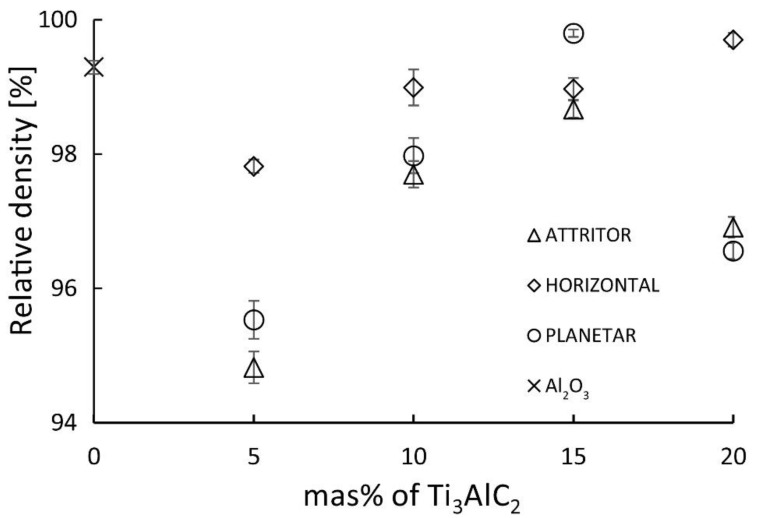
The relative density of Al_2_O_3_/Ti_3_AlC_2_ composites.

**Figure 3 materials-15-06909-f003:**
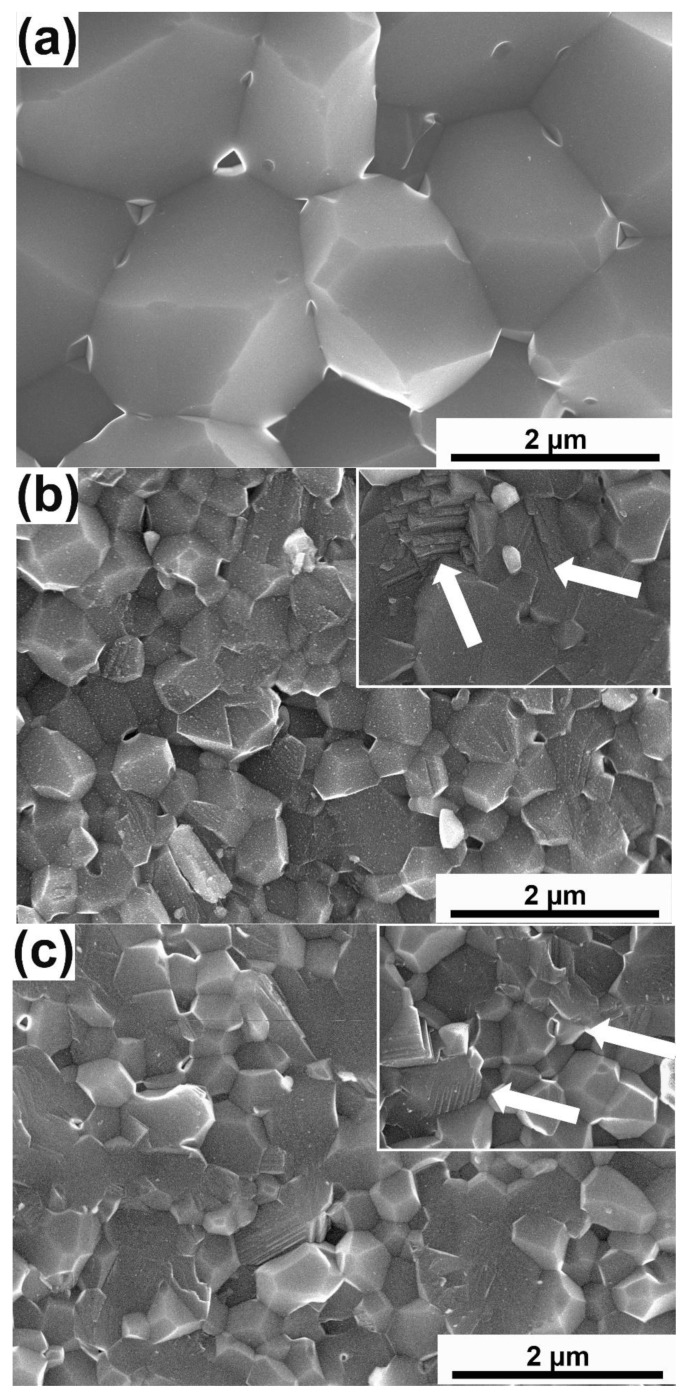
The fracture surface of (**a**) Al_2_O_3_, (**b**) Al_2_O_3_—10 wt% of Ti_3_AlC_2_ from horizontal mill, (**c**) Al_2_O_3_—10 wt% of Ti_3_AlC_2_ from Atrittor mill, (**d**) Al_2_O_3_—10 wt% of Ti_3_AlC_2_ from planetary mill (the arrows mark areas with a variable fracture mechanism).

**Figure 4 materials-15-06909-f004:**
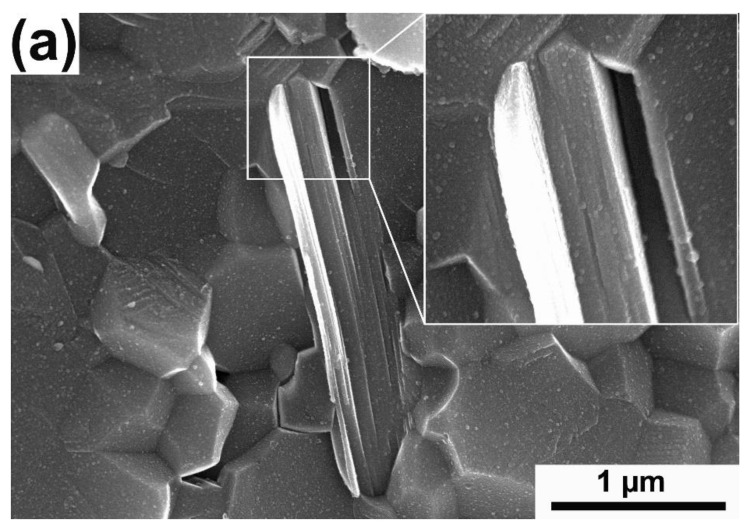
The microstructure of (**a**) Al_2_O_3_—5 wt% of Ti_3_AlC_2_ from planetary mill, (**b**) Al_2_O_3_—5 wt% of Ti_3_AlC_2_ from Atrittor mill (**c**) Al_2_O_3_—5 wt% of Ti_3_AlC_2_ from horizontal mill (the arrows mark the porosity on the interface).

**Figure 5 materials-15-06909-f005:**
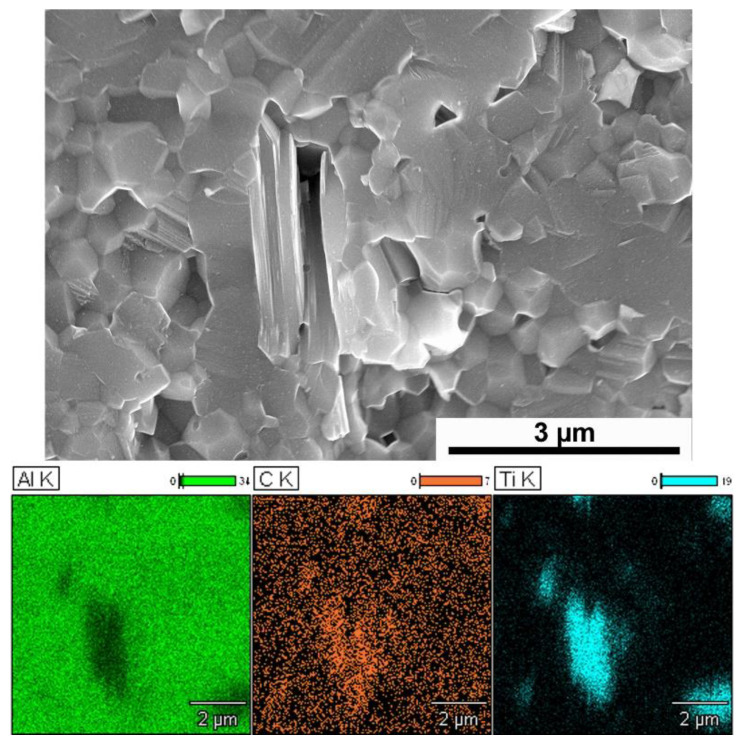
The EDS (Energy-dispersive X-ray spectroscopy) elemental map of Al_2_O_3_—5wt% of Ti_3_AlC_2_ composites.

**Figure 6 materials-15-06909-f006:**
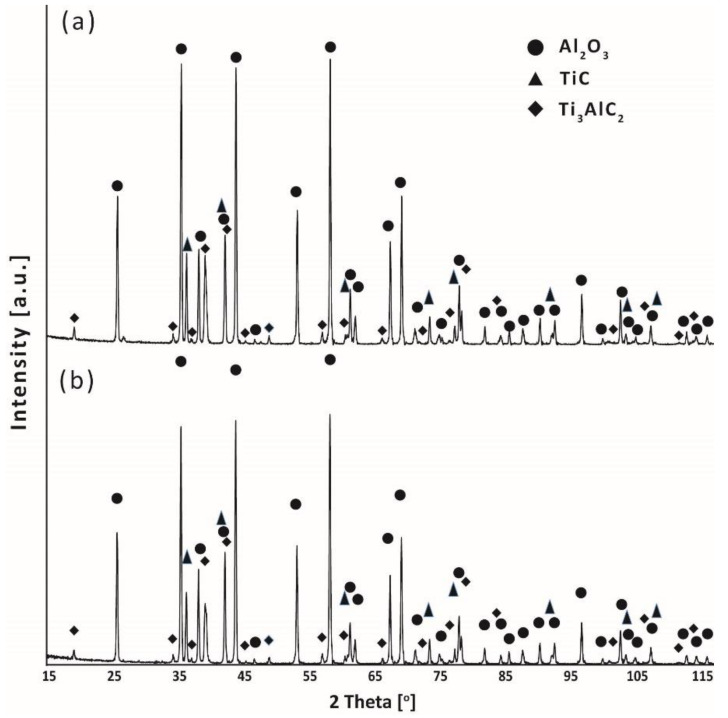
XRD analysis of (**a**) Al_2_O_3_ + 20 wt% of Ti_3_AlC_2_ addition from horizontal mill, (**b**) Al_2_O_3_ + 20 wt% of Ti_3_AlC_2_ addition from planetary mill.

**Figure 7 materials-15-06909-f007:**
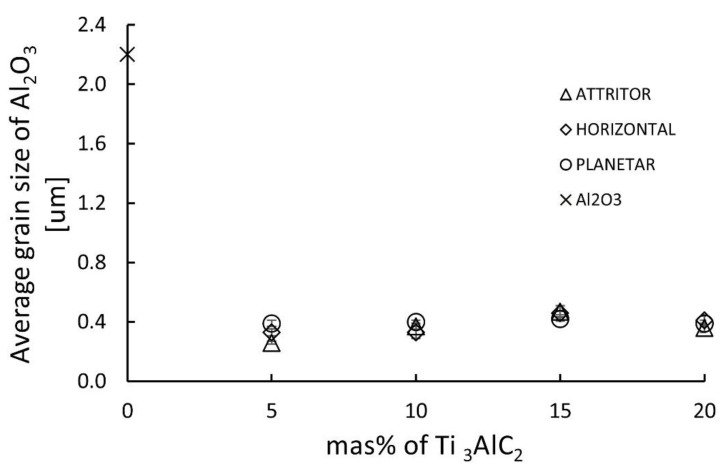
Average grain size of alumina in Al_2_O_3_-Ti_3_AlC_2_ composites.

**Figure 8 materials-15-06909-f008:**
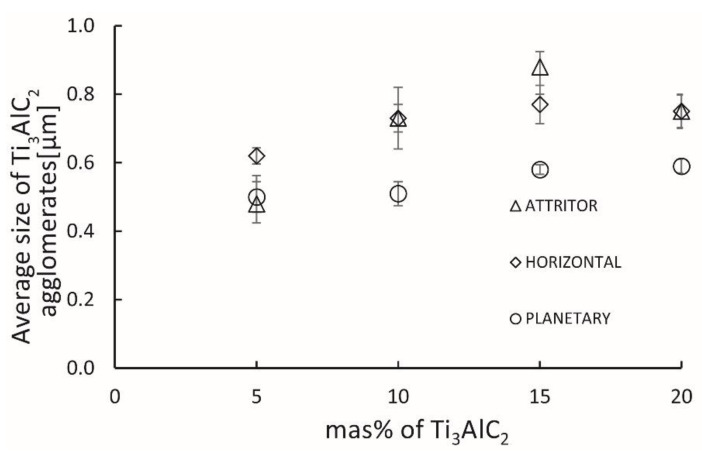
Average grain size of Ti_3_AlC_2_ agglomerates in Al_2_O_3_-Ti_3_AlC_2_ composites.

**Figure 9 materials-15-06909-f009:**
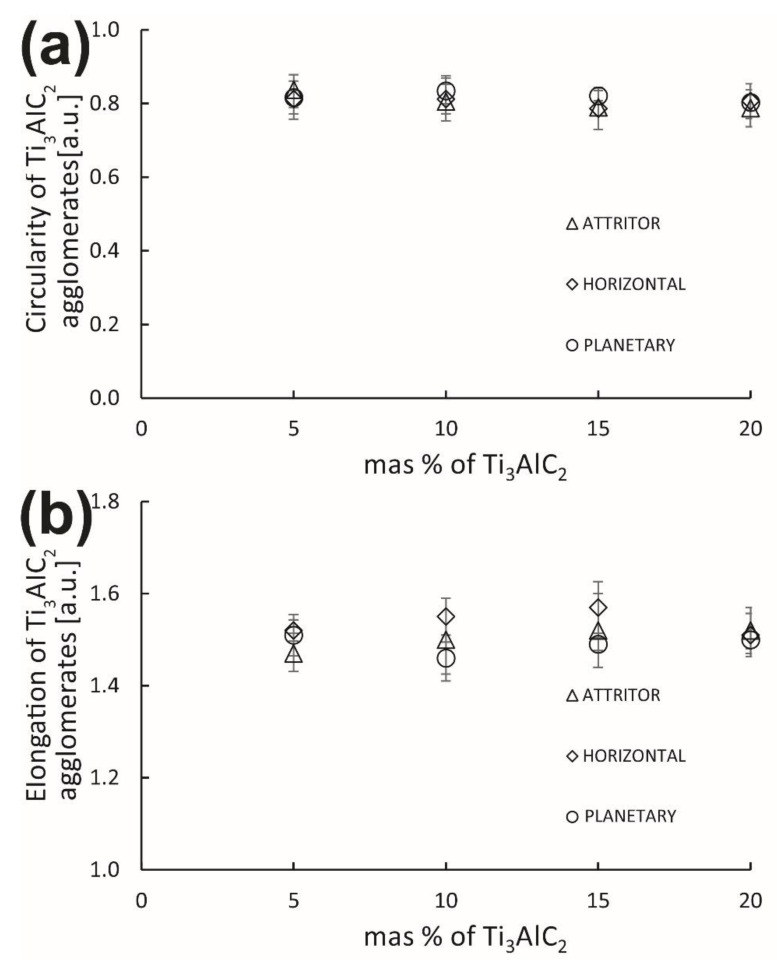
The (**a**) circularity and (**b**) elongation of Ti_3_AlC_2_ agglomerates.

**Figure 10 materials-15-06909-f010:**
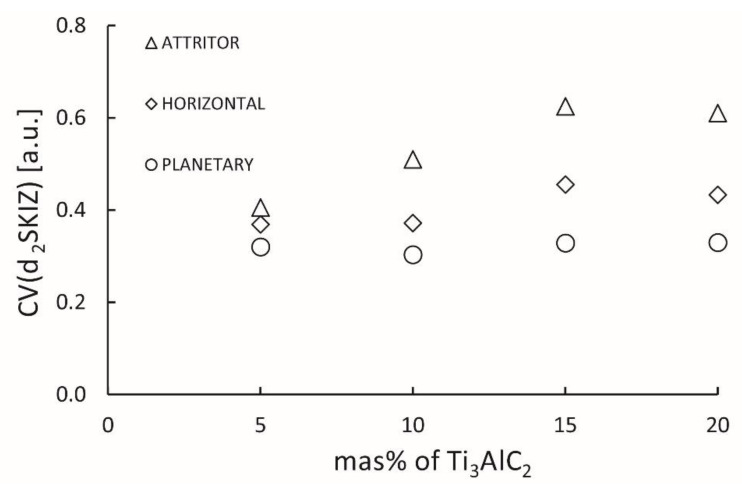
The CV_SKIZ_ (coefficient of SKIZ variation) of homogeneity of the microstructure of obtained composites.

**Figure 11 materials-15-06909-f011:**
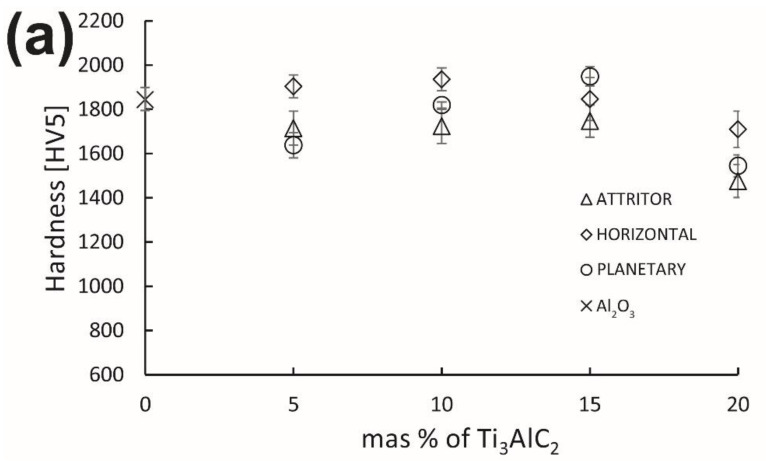
(**a**) Hardness of Al_2_O_3_-Ti_3_AlC_2_ composites; (**b**) fracture toughness of Al_2_O_3_-Ti_3_AlC_2_ composites.

**Table 1 materials-15-06909-t001:** The parameters of used powders. APS—average particle size.

Powder	Purity	APS *	Manufacturer
Titanium	99.6%	<20 µm	GoodFellow, Cambridge, UK
Aluminum	99.7%	6.74 µm	Bend-Lutz Co. Skawina, Poland
Synthetic graphite	99.9%	<20 µm	Sigma-Aldrich, St. Louis, MO, USA
Alumina	99.99%	0.125 nm	Taimei Chemicals Co., Ltd., Tokyo, Japan

* average particles size.

## Data Availability

All the data is available within the manuscript.

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
