# Peer review of "Novel Alumina Matrix Composites Reinforced with MAX Phases—Microstructure Analysis and Mechanical Properties"

_materials, 2022, doi:10.3390/ma15196909_

Round 1

Reviewer 1 Report

Petrus et al. have submissed a well-structured, consistent paper about MAX reinforced alumina composites. I have only a few minor questions and format remarks.

Major issue:

    Lines 320-322: this is not a result-supported conclusion of the paper. It should be moved into the introduction.

Minor questions:

    Lines 75-81: particle size of all of the raw materials was mentioned except for Al. What was the particle size of Al?

    Line 85: why the molar ratio of Ti:Al:C was 3:1:1.9 (instead of 3:1:2)?

    Line 98: what was the vacuum pressure?

    Lines 125-129: could you estimate the amount of different phases from the X-ray intensities? (Eg. with the RIR method.)

    Lines 150-151: " the highest amount of pores and voids occur in the composite obtained using a planetary mill." Could you quantify this?

    Lines 171-179 and figure 5.: what is the Ti:Al:C ratio within the "blue spot"? Does it confirm that is Ti3AlC2 instead of TiC?

    Lines 171-179 and figure 6.: could you estimate the amount of different phases from the X-ray intensities? (Eg. with the RIR method.)

Format remarks, typos:

    Section 4 is missing.
    Section 3 (results) should be divided by subsections.
    Lines 27, 28 and 98: unnecessary comma before 'and'/'or'.
    Lines 69 and 72: typo about subscript.
    Line 70: "Self-propagating High-temperature Synthesis" or "self-propagating high-temperature synthesis".
    Line 90: unit daN should be avoided.
    Line 109: CuK_{alpha_1}
    Figures 3. and 4.: caption should mention what are the white arrows standing for.
    Figure 5.: Al and C elemental map should have different color.
    Figures 2, 7, 8, 9 and 10: "mass%"
    Figure 8: subfigure b is hiding the xlabel of subfigure a.

Reviewer 2 Report

The paper investigated the manufacturing of alumina composites with the addition of MAX phases. The paper is interesting, and needs some improvements,

1. More recent related references should be added.

2. The lable of x axial in Fig.8a is not shown.

3. The conclusion should focus on the main findings.

Reviewer 3 Report

The authors investigated the effect of MAX phases (Ti3AlC2) on the microstructure and mechanical properties of alumina matrix composites. The presented results are interesting; thus, the manuscripts could be considered for publication in Materials. I have some comments and suggestions to improve the quality of the manuscript as follows:

1. Please explain why the density of composites decreases with the increase of Ti3AlC2 contents.

2. Relative density of the composite should be calculated and presented.

3. Distribution of grain size of the composites with different Ti3AlC2 contents should be calculated and included in the manusript.

4. Also provide the SEM images with the distribution of Ti3AlC2 and Ti3AlC2 clusters for the composites containing different Ti3AlC2 contents.

Round 2

Reviewer 3 Report

The manuscript could be accepted for publication in its current form.